# Analysis of Chemical Composition and Assessment of Cytotoxic, Antimicrobial, and Antioxidant Activities of the Essential Oil of *Meriandra dianthera* Growing in Saudi Arabia

**DOI:** 10.3390/molecules24142647

**Published:** 2019-07-22

**Authors:** Ramzi A. Mothana, Fahd A. Nasr, Jamal M. Khaled, Mohammed AL-Zharani, Omar M. Noman, Nael Abutaha, Adnan J. Al-Rehaily, Omar M. Almarfadi, Ashok Kumar, Mine Kurkcuoglu

**Affiliations:** 1Department of Pharmacognosy, College of Pharmacy, King Saud University, P.O. Box 2457, Riyadh 11451, Saudi Arabia; 2Medicinal Aromatic, and Poisonous Plants Research Center, Department of Pharmacognosy, College of Pharmacy, King Saud University, Riyadh 11451, Saudi Arabia; 3Department of Botany and Microbiology, College of Science, King Saud University, Riyadh 11451, Saudi Arabia; 4Biology Department, College of Science, Al Imam Mohammad Ibn Saud Islamic University (IMSIU), Riyadh 11451, Saudi Arabia; 5Bioproducts Research Chair, Department of Zoology, College of Science, King Saud University, Riyadh 11451, Saudi Arabia; 6Vitiligo Research Chair, College of Medicine, King Saud University, Riyadh 11451, Saudi Arabia; 7Department of Pharmacognosy, Faculty of Pharmacy, Anadolu University, 26470 Eskisehir, Turkey

**Keywords:** *Meriandra dianthera*, *Meriandra bengalensis*, volatile oil, GC-MS, anticancer, antimicrobial, antioxidative, apoptosis

## Abstract

The essential oil of *Meriandra dianthera* (Konig ex Roxb.) Benth. (Synonym: *Meriandra bengalensis*, Lamiaceae) collected from Saudi Arabia was studied utilizing GC and GC/MS. Forty four constituents were identified, representing 96.8% of the total oil. The *M. dianthera* essential oil (MDEO) was characterized by a high content of oxygenated monoterpenes (76.2%). Camphor (54.3%) was the major compound in MDEO followed by 1,8-cineole (12.2%) and camphene (10.4%). Moreover, MDEO was assessed for its cytotoxic, antimicrobial, and antioxidant activities. MDEO demonstrated an interesting cytotoxic activity against all cancer cell lines with IC_50_ values of 83.6 to 91.2 μg/mL, especially against MCF-7 cancer cells. Using labeling with annexin VFITC and/or propidium iodide (PI) dyes and flow cytometer analysis, the apoptosis induction was quantitatively confirmed for MCF-7 cells. The MDEO exhibited a considerable antimicrobial activity against all bacterial and fungal strains with minimum inhibitory concentration (MIC)-values of 0.07 to 1.25 mg/mL. The most sensitive microbial strain was *Staphylococcus aureus* (MIC: 0.07 mg/mL). Minimum bactericidal concentration (MBC) or minimum fungicidal concentration (MFC) values were determined one time higher than that of MIC’s. Additionally, the MDEO revealed a strong activity for reducing β-carotene bleaching with a total antioxidant value of 72.6% and significant DPPH free radical scavenging activity (78.4%) at the concentration 1000 μg/mL.

## 1. Introduction

The genus *Meriandra* (family: Lamiaceae) is represented by four species that are distributed throughout Asia, Africa, and India [1] (Wood, 1997). *Meriandra dianthera* (Roth ex Roem. & Schult.) Briq. [synonym: *Meriandra bengalensis* (Konig ex Roxb.) Benth.] is one of the members of this genus, which occurs as a perennial much-branched, erect, aromatic undershrub up to a high of 3–6 ft and grows in Saudi Arabia, Yemen, and Eritrea on rocky hills between 2000 and 2800 m [1,2]. *M. dianthera* is utilized traditionally in many places as an antiseptic, astringent, carminative, and antirheumatic agent [3,4,5]. The aerial part and the root of this plant species are widely used in Saudi and Yemeni folk medicine. The infusion of the plant is indicated for wounds as an antiseptic agent and as a remedy for urinary tract infections [5,6]. In Eritrean traditional medicine, *M. dianthera* is taken for hypertension, malaria, hepatitis, infections, and diabetes [7,8]. Former works on the leaves and the roots of this plant showed the presence of volatile oil, sesquiterpenoids, abietane diterpenoids, triterpenoids, and flavonoids [4,9,10,11]. In addition, previous study on the roots of *Meriandra dianthera* (*Meriandra bengalensis*) collected from Yemen have shown interesting cytotoxic, antimicrobial, and antioxidative activities and led to the purification and the characterization of four abietane diterpenoids [11].

Reviewing the obtainable existing studies, nothing was found concerning the volatile oil of the aerial part of *M. dianthera* growing in Saudi Arabia. In continuation of our research on volatile oils and their potential anticancer, antimicrobial, and antioxidative activities, we describe here for the first time the chemical composition, the cytotoxic, the antimicrobial, and the antioxidant activities of the essential oil of *Meriandra dianthera* from Saudi Arabia.

## 2. Results

The *Meriandra dianthera* yielded 2.33% (*w/w*) of a colorless and fragrant essential oil.

### 2.1. Chemical Composition of the Essential Oil

The gas chromatography with flame ionization detector (GC/FID) and the GC/MS results are demonstrated in Table 1. It demonstrated the chemical content of the investigated oil, the retention indices, the percentages, and the identification methods. The identified compounds are represented in order of their elution on the HP Innowax column. A total of 44 compounds, representing 89.2% of the total oil, could be identified. *M. dianthera* essential oil (MDEO) was characterized by a high content of oxygenated monoterpenes (76.2%). Camphor (54.3%) was the major constituent in the volatile oil followed by 1,8-cineole (12.2%), camphene (10.4%), and borneol (3.1%) (Table 1, Figure 1).

### 2.2. Cytotoxicity and Apoptosis Staining

As displayed in Table 2 and Figure 2, *Meriandra dianthera* essential oil (MDEO) demonstrated an interesting cytotoxic activity against all cancer cell lines with IC_50_ values ranging between 83.6 to 91.2 μg/mL. The highest effect was found against Human cancer cell lines MCF7 (breast cancer) and LOVO (colon cancer). The cytotoxic effect on the normal human umbilical vein endothelial cell line (HUVEC) was obviously lower than on the cancer cells (Table 2 and Figure 2).

As shown in Figure 3, control cells retained their cellular morphology and confluency and were oval, intact, and uniformly stained. In contrast, treated cells with MDEO showed strong effects on cell morphology as was obvious from the densities decrease of the cells, the shrinkage detachment, and the loss of cell integrity. The typical marks of apoptosis were also evident in the treated cells (Figure 3).

The apoptosis induction was confirmed quantitatively using flow cytometer analysis. The cells were labeled with Annexin VFITC and/or propidium iodide (PI) dyes (Figure 4). It was shown that the cells were split into four quadrates based on the cell stages: viable, early apoptosis, late apoptosis, and necrosis quadrates. At 24 h of treatment, MDEO induced apoptosis to the treated MCF-7 cell lines. There was no apoptosis (0%) in the untreated cells, indicating that the treatment with the essential oil caused induction of apoptosis. As the time of treatment increased to 24 h, the percentage of apoptotic cells (early and late) also increased proportionally to time of exposure.

### 2.3. Antimicrobial Activity

Minimum inhibitory concentrations (MICs), minimum bactericidal concentrations (MBCs), and minimum fungicidal concentrations (MFCs) of MDEO are displayed in Table 3. As demonstrated in Table 3, the volatile oil possessed variable degrees of growth inhibition of the bacterial and the fungal strains with MIC-values ranging between 0.07 and 1.25 mg/mL. The most sensitive microbial strain was the Gram-positive *Staphylococcus aureus* (MIC: 0.07 mg/mL). MBC or MFC values were always about twofold higher than that of MIC’s (Table 3).

### 2.4. Antioxidant Activity

The antioxidative activity and the radical scavenging results of MDEO are demonstrated in Table 4. In the β-carotene-bleaching test, MDEO displayed a great capacity to reduce the β-carotene bleaching at a concentration of 1000 μg/mL with a total antioxidant value of 72% (Table 4). Furthermore, at the highest concentrations 500 and 1000 μg/mL, MDEO possessed significant radical scavenging activity (72 and 78%) (Table 4).

## 3. Discussion

Studies on medicinal plants with anticancer and antimicrobial activities need to be continued in order to find novel, more effective, and affordable medications. With this aim, this investigation searched for important and promising natural products from Saudi medicinal plants. In this study, we investigated the chemical constitution of the volatile oil of *M*. *dianthera* (*M. bengalensis*) by utilizing GC/FID and GC/MS. Additionally, we explored in the present investigation anticancer, antimicrobial, and antioxidant activities as well as the possible apoptotic induction using Annexin VFITC and/or PI dyes and flow cytometer analysis.

Apparently, this is the first study on chemical content, cytotoxic, antimicrobial, and antioxidative activities of the essential oil of *M. dianthera* grown in Saudi Arabia. Generally, the current information about the genus *Meriandera* is limited. Comparing our data with antecedently published knowledge on chemical content of the volatile oil of *M. dianthera* grown in other places uncovered few substantial quantitative and qualitative variations between them.

Our data showed that MDEO displays a high content of oxygenated monoterpenes (76.2%), among which camphor (54.3%) is the predominant constituent followed by 1,8 cineole (12.2%) (Table 1). This is, to some extent, in agreement with an earlier study on the essential oil of *M. dianthera* grown in Yemen [4], which revealed the presence of only 12 compounds, and among them, camphor (43.6%) and 1,8 cineole (10.7%) were the major constituents. Our results are partly in agreement with a recent study done by Sium and co-workers in 2017 [8] who investigated the chemical content and the anti-diabetic activity of the methanol extract of *M. dianthera* grown in Eritrea and revealed a predominance of camphor, borneol, α-terpineol, and β-eudesmol as well as a significant anti-diabetic activity through α-glucosidase and α-amylase inhibition activities [8]. Unlike our results, palmitic acid methyl ester and α-linolenic acid methyl ester were characterized in *M. dianthera* by Sium and co-workers in 2017 [8]. Additionally, Bruno et al. (1985) [12] asserted the predominance of camphor (80%) in the volatile oil of *M. dianthera* grown in Italy. However, our results are not quite in agreement with a former report on the volatile oil of *M. dianthera* grown in India, which indicated linalool (68.4%) as a major component [13].

Of course, the divergence in the content of the essential oil of the plant species including *M. dianthera* may be ascribed to various geographical and environmental aspects, for example, weather conditions, stage of development, altitude above sea level atmosphere, type of soil, time of harvest, and extraction methods of the plants, and thus influences greatly the chemical content of the oils [14].

In the present investigation, promising cytotoxic, antimicrobial, and antioxidative activities were noticed for MDEO. The cytotoxic effect of MDEO was assessed against three types of cancer cell lines (MCF-7, HepG2, and LoVo) utilizing MTT tests. MDEO displayed noteworthy cytotoxic activity against the three investigated cancers. In general, literature data on MDEO cytotoxicity are still scarce. In agreement with our results, a work presented by Ali et al. (2012) [4] highlighted a strong cytotoxic activity of *M. dianthera* (*M. bengalensis*) essential oil from Yemen (71% at 100 µg/mL) on human colon tumor HT-29 cells.

Previous studies showed that dominated compounds, e.g., borneol, α-pinene, and β-caryophyllene, possess in vitro cytotoxicity to various cancer cells such as MCF-7, MDA-MB-468, HepG2, and UACC-257 cancer cell lines. Consequently, the cytotoxic activity of MDEO could supposedly be ascribed to constituents that were identified as predominant constituents, e.g., camphor, 1,8-cineol, borneol, and α-pinene [15,16,17].

Apoptosis or programmed cell death is a profoundly organized physiological process to dispose of harmed or abnormal cells. Therefore, the promotion of apoptosis in tumor cells is considered exceptionally valuable in the therapy as well as in the avoidance of cancer. Apoptotic cells display a characteristic morphology and particular molecular features. Induction of apoptosis in tumor cells or cancer tissues is recognized as an effective technique in anticancer chemotherapy. Moreover, the capacity to induce apoptosis is a solid marker for assessing potential agents for the treatment of cancer [18,19].

In the current work, the morphology of apoptotic cells was observed utilizing fluorescence microscopy after double staining with acridine orange and ethidium bromide (AO/EtBr) as well as DAPI fluorescence. Additionally, the apoptosis was confirmed by flow cytometry after staining with annexin V-FITC/PI. Annexin V-FITC/PI binds to Phosphatidylserine (PS), which translocates from the inner face of the plasma membrane to the cell surface in early stage apoptosis and can be visualized by the green fluorescence of the chromophore used (FITC/PI) [20,21]. DAPI dye was utilized for counterstaining the DNA of apoptotic cells with fluorescence blue color [22]. Intensity of DAPI labeling represented the status of chromatin condensation. The pale uniform DAPI color pointed out an ordinary and uncondensed chromatin with an oval nucleus. Brighter DAPI color uncovered a condensed chromatin, which was patchy in shape. These are the indications of apoptosis [20,23]. The obtained results suggest that MDEO can induce apoptosis in MCF-7 cells.

A previous study by Moteki and co-workers (2002) [24] revealed a specific induction of apoptosis by 1,8-cineole in human leukemia Molt 4B and HL-60 cells. It was found that morphological alterations showing apoptotic bodies and fragmentations of DNA were observed in the human leukemia HL-60 cells treated with 1,8-cineole. Consequently, it is conceivable that the induction of apoptosis in MCF-7 by MDEO is ascribed to the high percentage of 1,8-cineole in MDEO.

Little has been noted on the antimicrobial activity of the *M. dianthera* essential oil. Only one report on the antimicrobial activity of *M. dianthera* essential oil was found [4]. Our outcomes were not in agreement with the data reported by Ali and co-workers, who demonstrated only a weak antimicrobial activity against all tested microorganisms. It was shown that the Gram-positive *S. aureus* appeared to be the most resistant one [4], while in our investigation, *S. aureus* was the most sensitive microbial strain. Some former investigations bolster the hypothesis that particular essential oil constituents, e.g., camphor, 1,8-cineole, camphene, and borneol produced from various plant species, are responsible for antimicrobial activity [25,26,27,28,29,30].

## 4. Materials and Methods

### 4.1. Plant Material

The aerial part of *Meriandra dianthera* was collected from the Aqabat Tanoma in Saudi Arabia in February 2016. The plant was authenticated at the Pharmacognosy Department, College of Pharmacy, King Saud University (KSU, Riyadh, SA), and a voucher sample (KSU 16386) was deposited in the herbarium at the Pharmacognosy Department, College of Pharmacy, King Saud University, Riyadh, Saudi Arabia.

### 4.2. Preparation of the Volatile Oil

The collected plant was dried under shade at room temperature and powdered. The ground aerial part of *Meriandra dianthera* (300 g plant material in 1 L water) was submitted for 3 h to hydrodistillation utilizing a Clevenger-type apparatus. The obtained volatile oil was dried over anhydrous sodium sulphate. After that, the oil was filtered and kept at +4 °C until tested and analyzed.

### 4.3. Gas Chromatography/Mass Spectrometry Analysis

Gas chromatographic investigation was performed on a 5975 Gas Chromatograph coupled with mass spectrometer (Agilent, USA; SEM Ltd., Istanbul, Turkey). As a stationary phase, Innowax FSC column (60 m × 0.25 mm, 0.25 μm film thickness) was used, while helium was the mobile phase (0.8 mL/min). The volume injected was 0.1 µL with a split ratio of 40:1. The oven temperature of the GC was at first fixed at 60 °C for 10 min, then increased to 220 °C at a rate of 4 °C/min, held constant for 10 min and thereafter increased to 240 °C at a rate of 1 °C/min. The injector and the transfer line temperatures were fixed at 250 and 280 °C, respectively. MS detection was carried out at 70 eV with scan mass range *m/z* 35–450.

### 4.4. Gas Chromatography/FID Analysis

The investigation was implemented on an Agilent Technologies 6890 N GC system with flame ionization detector. The temperature of the FID was set at 300 °C. The same column utilized in GC/MS tests as well as the same operational conditions were implemented to a triplicate. Simultaneous auto injection was carried out to get equivalent retention times. The quantification (relative proportions) of the recognized constituents was calculated from the FID peak area percent normalization.

### 4.5. Identification of Compounds

The essential oil constituents were identified by comparing the mass spectra with those of similar compounds in the Adams Library [31], the Mass Finder Terpenoid Library [32], the Wiley GC/MS Library [33], and our own Baser Library of Volatile Oil Constituents on the basis of their retention indices. The identification was achieved by comparing the retention times with authentic reference standards and by comparing the retention index (RRI) relative to C_8_–C_30_ of *n*-alkanes under the same above mentioned operating conditions [34]. The results are demonstrated in Table 1 as mean percentage ± standard deviation (SD) (*n* = 3).

### 4.6. Determination of Anticancer Activity on Human Cancer Cell Lines

#### 4.6.1. Cancer Cell Lines and Culture

Three different human cancer cell lines, breast (MCF-7), liver (HepG2), and colon (LoVo) cells were developed in Dulbecco’s Modified Eagle Media (DMEM) supplemented with 2 mM l-glutamine, 10% fetal calf serum, and 1% penicillin-streptomycin.

#### 4.6.2. MTT Assay

Three different human cancer cell lines, breast (MCF-7), liver (HepG2), colon (LoVo), and one normal human umbilical vein endothelial (HUVEC) cell were developed in DMEM supplemented with 10% fetal bovine serum (FBS) and incubated at 37 °C with 5% CO_2_. The cytotoxic activity of MDEO was determined using MTT assay [35]. Briefly, one mL of cell suspension (5 × 10^4^ cells/mL) was seeded in a 24-well plate; afterwards, cells were treated for 24 h with various concentrations of MDEO. One hundred microliters of 5 mg/mL MTT solution was added to all wells, and plates were incubated at 37 °C for 2–4 h. The reduced MTT was measured at 540 nm with microplate ELISA reader (Thermo Fisher Scientific, Waltham, MA, USA). Wells with untreated cells were considered as controls. Vinblastine was used as a positive control. For each compound tested, the IC_50_ (concentration of tested compound needed to inhibit cell growth by 50%) was calculated from the dose–response curves using the following formula:% Cell Viability = Mean absorbance of treated sample/Mean absorbance of control × 100

#### 4.6.3. Light Microscopy

MCF-7 cells were grown in 12-well plates and incubated for 24 h with and without MDEO at concentrations of IC50. The morphological changes of the apoptotic cells were observed using a phase contrast inverted microscope (EVOS^®^ FL Color, Life Technologies, Carlsbad, CA, USA).

#### 4.6.4. Apoptosis Assessment by DAPI Staining and Acridine Orange/Ethidium Bromide Assays

DAPI staining was performed using MCF-7 cells that were cultured in a 12-well tissue culture grade plate (Nest, Wuxi, China) for 24 h. After incubation with IC_50_ for 24 h, cells were washed in phosphate buffered saline (PBS) and fixed with ethanol for 15 min at room temperature. Cells after washing with PBS were stained with DAPI (2 μg/mL) and incubated in the dark for 30 min. The cells were then examined and imaged using a fluorescence microscope (EVOS^®^ FL Color, Life Technologies, Carlsbad, CA, USA).

For acridine orange/ethidium bromide assay, the amount of 2 µL of acridine orange/ethidium bromide (one part each of 3 mg/mL acridine orange and 3 mg/mL ethidium bromide in PBS) was mixed with 1 ml cell suspension in a 12-well plate. Cells were examined by EVOS^®^ imaging connected to a digital imaging system.

#### 4.6.5. Flow Cytometry Analysis of Cell Apoptosis

Apoptosis detection was performed in accordance with the method described in the manufacturer’s instructions of Alexa Fluor 488 Annexin V/Dead Cell Apoptosis kit (Thermo Fisher Scientific, Inc. Waltham, MA, USA). In brief, MCF-7 cells were seeded on tissue culture plate (6-well) at a density of 5 × 10^4^ cells/well for 24 h at 37 °C for adherence and were then treated with IC_50_ of MDEO. Following a 24 h treatment, the cells were collected, washed twice in cold PBS, and resuspended in kit specific binding buffer. Then, the cells were incubated in the dark with 5 μL of Annexin V-FITC and 5 µL of PI for 15 min. After incubation, 400 µL of annexin-binding buffer was added, and the samples were immediately analyzed by flow cytometry.

### 4.7. Determination of Antimicrobial Activity

#### 4.7.1. Test Microorganisms

The microbial strains applied in this investigation were *Staphylococcus aureus* (ATCC 25923), *Staphylococcus epidermidis* (ATCC 12228), *Escherichia coli* (ATCC 25922), *Acintobacter* sp. (ATCC 49139) wild strain, *Aspergillus ochraceus* (AUMC 9478), *Penicillium chrysogenum* (AUMC 9476), *Candida albicans* (ATCC 60193), and *Rhodotorula* sp. wild type.

#### 4.7.2. Minimal Inhibitory Concentrations (MIC)

The MIC values of *M*. *dianthera* essential oil (MDEO) against two Gram-positive bacteria, two Gram-negative bacteria, and four fungi were evaluated using a microwell dilution assay as reported earlier [36] with modifications. With sterile round-bottom 96-well plates, duplicate two-fold serial dilutions of MDEO (100 µL/well) were made ready in the proper broth (Mueller–Hinton or Sabouraud dextrose broth) containing 5% (*v/v*) DMSO to set up a range of concentrations (20 to 0.156 µL/mL) of MDEO. Then, 100 µL (1 × 10^6^ CFU/mL) of the bacterial or the fungal suspension that was formerly prepared in suitable broth was given in each well except those in columns 10, 11, and 12, which were utilized as negative controls saline and media sterility. The last well in the plates was served for microbial growth without MDEO. Then, the 96-well plates were incubated at the suitable temperature for each strain for 24 h. The MIC of MDEO was specified as the lowest concentration showing no perceptible microbial growth. Gentamycin and nystatin (125 to 0.97 µg/mL) were used as positive controls. MBC and MFC values (minimal bactericidal concentration and minimal fungicidal concentration) were determined by taking a part of the liquid (5 µL) from each well that showed no growth and incubating on agar plates at 37 °C for another 24 h. The lowest concentration that disclosed no visible growth of bacteria or fungi was deemed as MBC or MFC.

### 4.8. Determination of Antioxidant Activity

#### 4.8.1. DPPH Radical-Scavenging Activity

The antioxidative activity of MDEO was estimated utilizing 2,2-diphenyl-1-picrylhydrazyl (DPPH) as reported earlier by Brand-Williams et al. (1995) [37]. This test evaluates the radical scavenging activity of the DPPH by the studied sample. Five concentrations of MDEO (10, 50, 100, 500, and 1000 μg/mL) were made ready. Then, 500 μL of MDEO was added to 125 μL DPPH methanol solution (1 mM) and 375 μL methanol and incubated for 30 min at room temperature within the dim. After that, the anti-DPPH activity was calculated by reading the absorbance at λ = 517 nm utilizing a UV-spectrophotometer (UV mini-1240, Shimadzu, Kyoto, Japan) and calculated applying the following formula:% of anti-radical activity = Abscontrol − Abssample/Abscontrol × 100

#### 4.8.2. β-Carotene Bleaching Test

The antioxidative activity of the *M. dianthera* essential oil (MDEO) was assessed by applying the β-carotene bleaching assay as reported earlier by Mothana et al. (2012) [38] with modification. β-carotene solution (1 mL of 200 μg in 1 mL chloroform) was given to a flask containing a solution of 200 μL of Tween-20 and 20 μL of linoleic acid. The chloroform was evaporated utilizing a rotatory evaporator, then 100 mL of distilled water was added, and the mixture was powerfully shaken for 2 min. Then, 200 μL of the MDEO (1000 μg/mL) was combined with 2 mL of the β-carotene-linoleic acid emulsion and incubated at 40 °C for 2 h. At the end, the absorbance was read at 470 nm at 30 min intervals. Rutin (1000 μg/mL) was applied as a positive control. The antioxidative activity was determined by the following formula: Antioxidant activity (%) = (Abs0 − Abst)/(Abs°0 − Abs°t) × 100
where Abs0 and Abs°0 are the absorbencies measured at zero time of incubation for the essential oil and the blank samples, respectively. Abst and Abs°t are the absorbencies for essential oil and the blank samples at 120 min, respectively.

### 4.9. Statistical Analysis

Results are displayed as means ± standard deviations (SD) for investigations performed in triplicate. The data were analyzed by one-way ANOVA applying Tukey test (IBM, SPSS, statistics 25). Significance difference was appointed by probability values of *p* ≤ 0.05.

## 5. Conclusions

To our knowledge, the present investigation determined the chemical content of the essential oil of *M. dianthera* (MDEO) grown in Saudi Arabia and its cytotoxic, antimicrobial, and antioxidant effects for the first time. The GC and the GC/MS investigations revealed that the chemical content was depicted by a high content of oxygenated monoterpenoids where camphor (54.3%), 1,8-cineol (12.2), and camphene (10.4%) were predominant. The results clearly showed that MDEO possess great cytotoxic, antimicrobial, and antioxidant activities. The obtained results suggest that MDEO can induce apoptosis in MCF-7 cells. Thus, our results support the hypothesis that MDEO could be a promising cytotoxic and antimicrobial agent. Additional investigations are necessary to ensure the mechanism of action of the cytotoxic and antimicrobial effects.

## Figures and Tables

**Figure 1 molecules-24-02647-f001:**
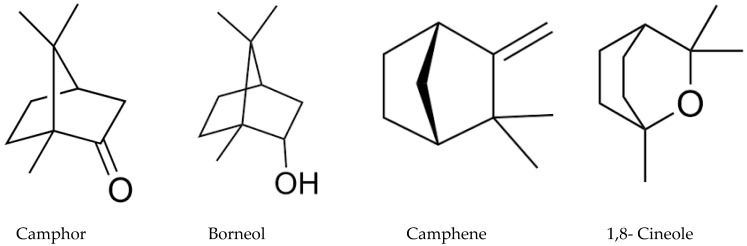
Major constituents in MDEO.

**Figure 2 molecules-24-02647-f002:**
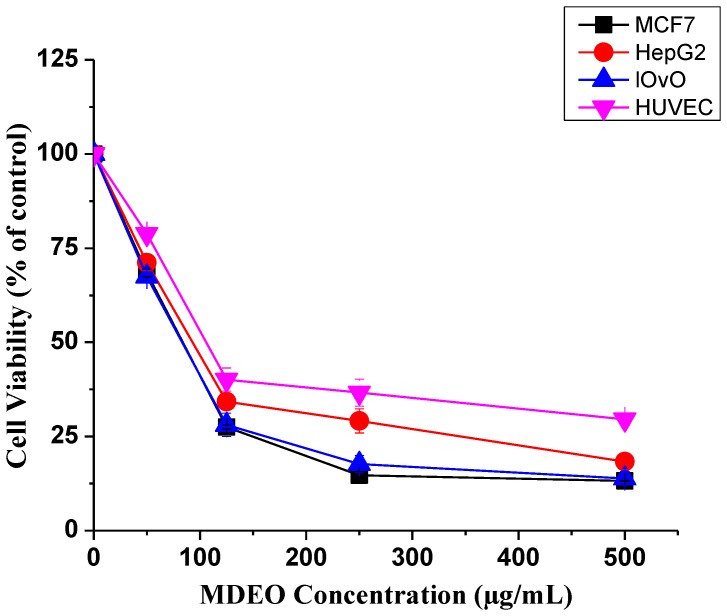
Cytotoxic effect of MDEO on different cancer cells. Dose-dependent curves treatment in MCF-7 (breast cancer), HepG2, LoVo (colon cancer), and human umbilical vein endothelial cell line (HUVEC) cells. Cells were cultured in 24-well plates and treated with different concentrations (50–500 µg/mL) for 48 h. Cell viability was measured by MTT assay. Data represent the mean ± S.D of three independent experiments carried out in triplicates.

**Figure 3 molecules-24-02647-f003:**
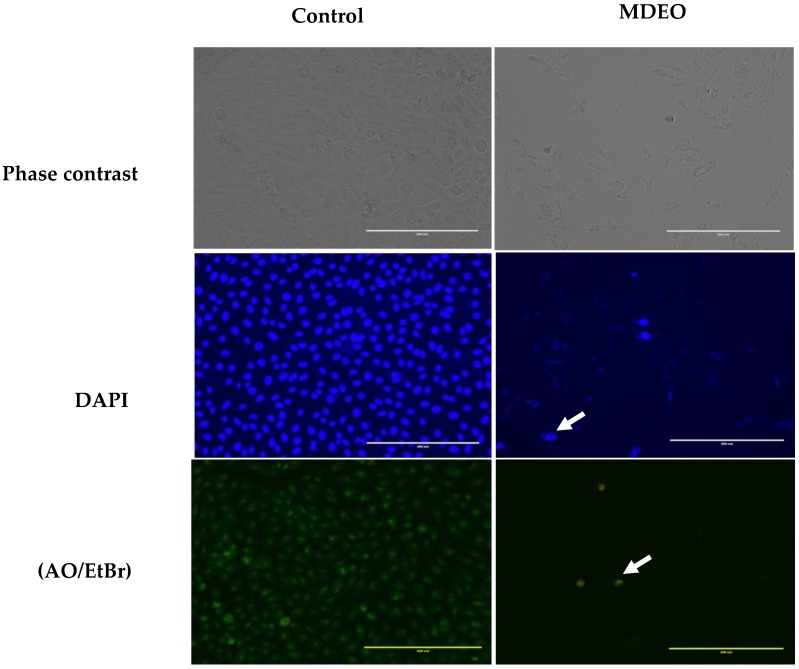
Detection of apoptosis by phase contrast and fluorescence microscopy in cells after treatment with MDEO. MCF-7 cells were treated with IC_50_ for 24 h. Cells were stained with DAPI, acridine orange, and ethidium bromide (AO/EtBr).

**Figure 4 molecules-24-02647-f004:**
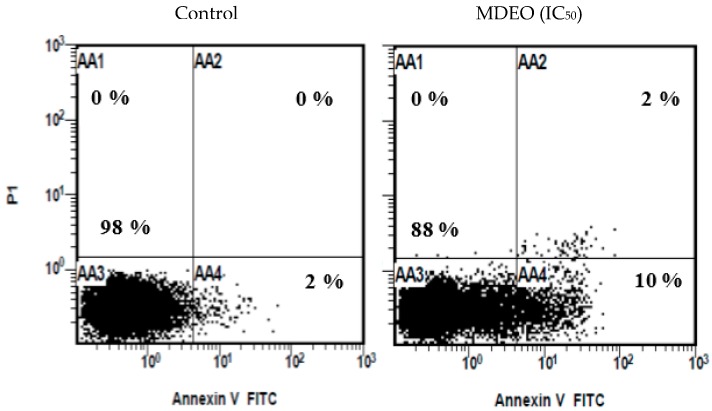
MDEO induces apoptosis in MCF-7 cells. AA1, AA2, AA3, and AA4 indicate the percentage of dead, late apoptotic, live, and early apoptotic cells, respectively.

**Table 1 molecules-24-02647-t001:** Chemical content of the essential oil of *Meriandra dianthera*.

No.	Compounds	RRI	%	Identification
1	Tricyclene	1014	0.4 ± 0.0	MS
2	α-Pinene	1032	2.5 ± 0.0	t_R_, MS
3	**Camphene**	1076	**10.4** ± 0.1	t_R_, MS
4	β-Pinene	1118	1.3 ± 0.0	t_R_, MS
5	Sabinene	1132	0.2 ± 0.0	t_R_, MS
6	Thuja-2,4 (10)-dien	1138	0.1 ± 0.0	MS
7	δ-3-Carene	1159	0.4 ± 0.0	S
8	Limonene	1203	0.8 ± 0.0	t_R_, MS
9	**1,8-Cineole**	1213	**12.2** ± 0.0	t_R_, MS
10	γ-Terpinene	1255	0.1 ± 0.0	t_R_, MS
11	*m*-Cymene	1278	0.2 ± 0.0	MS
12	*p*-Cymene	1280	0.4 ± 0.0	t_R_, MS
13	Camphenilone	1474	0.2 ± 0.0	MS
14	*trans*-Sabinene hydrate	1474	tr	t_R_, MS
15	*cis*-Linalool oxide (*Furanoid*)	1479	tr	MS
16	α-Campholenal	1484	0.2 ± 0.1	MS
17	**Camphor**	1532	**54.3** ± 0.1	t_R_, MS
18	Dihydroachillene	1544	0.2 ± 0.0	ms
19	Linalool	1553	0.2 ± 0.0	t_R_, MS
20	*cis*-Sabinene hydrate	1556	tr	t_R_, MS
21	Pinocarvone	1586	0.3 ± 0.0	MS
22	Bornyl acetate	1590	tr	t_R_, MS
23	Terpinen-4-ol	1611	0.7 ± 0.0	t_R_, MS
24	*trans*-Dihydrocarvone	1617	tr	t_R_
25	*trans-p*-Menth-2,8-dien-1-ol	1638	0.1 ± 0.0	MS
26	Myrtenal	1648	0.4 ± 0.0	MS
27	*cis*-Verbenol	1663	0.1 ± 0.0	MS
28	*trans*-Pinocarveol	1664	0.6 ± 0.0	s, MS
29	*cis-p*-Menth-2,8-dien-1-ol	1678	tr	MS
30	δ-Terpineol	1682	0.2 ± 0.0	t_R_, MS
31	*trans*-Verbenol	1683	1.1 ± 0.1	MS
32	α-Terpineol	1706	0.5 ± 0.0	t_R_, MS
33	Borneol	1719	3.1 ± 0.1	t_R_, MS
34	*p*-Mentha-1,5-dien-8-ol	1747	0.4 ± 0.0	MS
35	Myrtenol	1797	0.3 ± 0.0	MS
36	*trans*-Carveol	1845	0.3 ± 0.0	t_R_, MS
37	*m*-Cymen-8-ol	1849	0.5 ± 0.0	MS
38	*p*-Cymen-8-ol	1864	0.5 ± 0.1	t_R_, MS
39	Caryophyllene oxide	2008	0.9 ± 0.0	t_R_, MS
40	Ledol	2057	0.3 ± 0.0	MS
41	Guaiol	2104	0.2 ± 0.0	MS
42	Bulnesol	2232	0.5 ± 0.0	MS
43	β-Eudesmol	2255	1.1 ± 0.0	MS
44	Intermedeol	2260	0.2 ± 0.0	MS
	Monoterpene hydrocarbons		**17.4**	
	Oxygenated monoterpenes		**76.2**	
	Oxygenated sesquiterpenes		**3.2**	
	**Total**		**96.8**	

RRI, relative retention indices calculated against n-alkanes. %, calculated from the flame ionization detector (FID) chromatograms; tr, trace (<0.1%). Identification method: t_R_, identification based on the retention times (t_R_) of genuine compounds on the HP Innowax column; MS, identified on the basis of computer matching of the mass spectra with those of the Wiley and MassFinder libraries and compared with literature data.

**Table 2 molecules-24-02647-t002:** Cytotoxic activity of *Meriandra dianthera* essential oil (MDEO) on three cancer cells.

Sample	Cell Lines and IC_50_ (µg/mL)			
	MCF-7	HepG2	LoVo	HUVEC
MDEO	83.6 ± 0.6	91.2 ± 2.0	84.2 ± 1.0	105.7 ± 2.1
Vinblastine	2.2 ± 0.2	2.3 ± 0.48	1.5 ± 0.2	8 ± 1.6

**Table 3 molecules-24-02647-t003:** Antimicrobial activity of the MDEO (mg/mL).

Microorganisms		Activity	MDEO	Gentamycin	Nystatin
Bacteria	*S. aureus*	MIC	0.07	7.8	NT
		MBC	0.15	15.6	NT
	*S. epidermidis*	MIC	0.31	7.8	NT
		MBC	0.62	15.6	NT
	*E. coli*	MIC	1.25	3.9	NT
		MBC	2.5	7.8	NT
	*Acintobacter* sp.	MIC	1.25	3.9	NT
		MBC	2.5	7.8	NT
Fungi	*C. albicans*	MIC	0.31	NT	3.5
		MFC	0.62	NT	7.0
	*Rhodotorula* sp.	MIC	0.275	NT	3.5
		MFC	0.55	NT	7.0
	*A. ochraceus*	MIC	0.62	NT	3.5
		MFC	1.25	NT	7.0
	*P. chrysogenum*	MIC	0.31	NT	3.5
		MFC	0.62	NT	7.0

*S. aureus*: *Staphylococcus aureus* ATCC 25923, *S. epidermidis*: *Staphylococcus epidermidis* ATCC 12228, *E. coli*: *Escherichia coli* ATCC 25922, *Acintobacter* sp. ATCC 49139, *C. albicans*: *Candida albicans* ATCC 60193, *Rhodotorula* sp. Wild type, *A. ochraceus*: *Aspergillus ochraceus* AUMC 9478 and *P. chrysogenum*: *Penicillium chrysogenum* AUMC 9476, Values are given as mg/mL for MDEO and μg/mL for standard antibiotics, NT: Not tested. MIC: minimum inhibitory concentrations, MBC: minimum bactericidal concentration, and MFC: minimum fungicidal concentration.

**Table 4 molecules-24-02647-t004:** Antioxidant activity and free radical scavenging activity of MDEO.

Samples	Total Antioxidant Activity in % (1000 μg/mL)	Free Radical Scavenging Activity in % (DPPH) Assay				
		10	50	100 (µg/mL)	500	1000
MDEO	72.6 ± 2.9	22.5 ± 3.2	36.1 ± 2.8	48.5 ± 3.2	72.2 ± 3.0	78.4 ± 4.1
Ascorbic acid	NT	78.5 ± 4.0	84.8 ± 2.7	89.9 ± 3.1	91.1 ± 3.5	91.9 ± 2.3
Rutin	90.0 ± 3.8	NT	NT	NT	NT	NT

NT: Not tested.

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
