# Peer review of "Analysis of Chemical Composition and Assessment of Cytotoxic, Antimicrobial, and Antioxidant Activities of the Essential Oil of Meriandra dianthera Growing in Saudi Arabia"

_molecules, 2019, doi:10.3390/molecules24142647_

Round 1

Reviewer 1 Report

Thank you for agree with my concerns. I really hope that in the near future the AA are able to carry out the expts required.

Reviewer 2 Report

The corrections were made.

Reviewer 3 Report

The paper molecules-558846 « Analysis of chemical composition and assessment of cytotoxic, antimicrobial, and antioxidant activities of the essential oil of Meriandra dianthera growing in Saudi Arabia » has been carefully revised according to the reviewers’ comments.

As it stands, it has been largely improved and thus is now ready for publication.

This manuscript is a resubmission of an earlier submission. The following is a list of the peer review reports and author responses from that submission.

Round 1

Reviewer 1 Report

The paper molecules-510419 « Analysis of chemical composition and assessment of cytotoxic, antimicrobial and antioxidant activities of the essential oil of Meriandra bergalensis growing in Saudi Arabia » by R.A. Monthana et al. provides new  interesting data on the components of the essential oil extracted from the aerial parts of this aromatic plant (M. bergalensis) of Saudi Arabia (MBEO), its antioxydant, antibacterial and antifungal and its cytotoxic activities on cancer cell lines such as MCF-7. Camphor (54.3%), 1,8-cineole (12,2%), camphene (10.4) and borneol (3.1%) are identified as the major compounds. The authors provide some evidence for the induction of antimicrobial effects an  cytotoxic effects on human cells in culture. Interestingly, the effects appear to be more pronounced on cancer than on normal cells. Moreover, MBEO is effective in β-carotene bleaching and DPPH free radical scavenging activity.

Thus, the paper is well presented and reflects genuine scientific work. However, unfortunately, in some parts it is lacking some precision and information (see General and specific comments). In its present form, it is not yet fully acceptable for publication. It should become fully acceptable when taking into account the reviewers’ comments.

General comments

1.     When talking about the cancer cell lines in chapter 2.2. it would give more impact to the results if the authors would shortly present them. (Human cancer cell lines MCF7 (breast cancer), LOVO (colon cancer) HepG2(liver cancer are compared to normal human umbilical vein endothelial (HUVEC) cells) ). It appears that with regard to cell viability changes, Hep62 cells are less sensitive to MBEO than the other two cancer cell lines. Is this statistically significative ? In other words, is the effect of MBEO specific for cancer cells and what cud be the reason for that?

2.     Figure 3 does not show necessarily apoptosis but mostly cell death.

3.     In Table 3, it looks as if  always the MBC or MBF values are values that are always about twice of those of the MIC values.  Is this just a question of the difference in methodology used ? The authors might want to comment on that.

The presentation is somewhat misleading when comparing mg/ml concentrations of MBEO with mg/ml concentrations for the standard drugs. The authors should make it very clear that MBEO (if the MBC and MIC values are statistically significant) may show an antibacterial actvity on Staphylococcus aureus that comes closest to that of the antibiotic gentamycin. What is so special with S. aureus ? Although somewhat reactive, all the other microbials appear to be very much more resistant to MBEO than to the standard test compounds. Does the growth medium used for MBEO treatment have a protective effect ?

4.     L210 : It is not clear to the reader what the authors mean with « increasingly viable medications ».

5.     Lines 243-247 : The reasoning is not clear and not precise enough : Apparently, these sentences refer to  borneol, α-pinene and β-caryophyllene  and references 14-16 but not much to the present work. borneol, α-pinene and β-caryophyllene were found in small amounts (3.1, 2.5 and 0%, respectively). The types of essential oil and concentrations that affected the different cancer cell lines are not stated. Were these essential oil compounds predominantly present ? Do they allow the supposition that they may be mainly responsible for the observed cytoxic effects ? Or, is this meant to be a general statement that essential oils, somewhat similar to the camphor and 1,8-cineol types, show cytotoxic effects on cancer cells ? In any case, it is difficult to conclude because couriously enough, essential oil components when applied on their own have been shown in many instances to be less effective than the whole complex mixture of components that compose essential oils. Apparently, between individual essential oil components interesting interactions exist that underlie the overall biological activity of the essential oils. The authors may want to include a comment on this.

6.     Lines 248- 266 concerning apoptosis induction by MBEO : The authors should keep in mind that apoptosis induction is a complicated matter involving several different molecular pathways including mitochondrial functions. Also, the approach using cytofluorimetry is not an very easy matter (see Adan A, Alizada G, Kiraz Y, Baran Y, Nalbant A. Flow cytometry: basic principles and applications. Crit Rev Biotechnol. 2017 Mar;37(2):163-176. doi:10.3109/07388551.2015.1128876. Epub 2016 Jan 14. Review.). Very importanly, the measure of apoptosis provides indications of the underlying mechanisms. In thus regard, the observations reported by the authors are still preliminary but very promising. Thus, if possible, more research on this woud be worthwhile (see, for even more sophisticated approaches, Shaulov-Rotem Y, Merquiol E, Weiss-Sadan T, Moshel O, Salpeter S, Shabat D,Kaschani F, Kaiser M, Blum G. A novel quenched fluorescent activity-based probe reveals caspase-3 activity in the endoplasmic reticulum during apoptosis. Chem Sci. 2016 Feb 1;7(2):1322-1337. doi: 10.1039/c5sc03207e.) Monian P, Jiang X. Clearing the final hurdles to mitochondrial apoptosis:regulation post cytochrome C release. Exp Oncol. 2012 Oct;34(3):185-91.). By the way, also investigations on the concentration dependent induction of apoptosis (measuring different parameters of apoptosis such as morphological changes, DNA fragmentation (TUNEL), annexin, cytochrome C release, caspase 3 induction…) could constitute an interesting exension of the present work.

7.     L377 : 5%DMSO  may have on its own already a potent anti- oxidative (hydroxyl) radical effect. A control for DMSO alone is missing ? (the use of ethanol may give less biased results).

8.     L376 : dilution of MBEO in broth may reduce the actual activity of MBEO. Did the authors check for that ? The growth medium may reduce the possibility of MBEO to interact with bacterial or fungal cells. (This could explain the observed relative resistance  of most microbials to MBEO).

9.      

Specific comments

L34 : , especially

L36 : quantitatively confirmed for MCF-7 cells.

L40 : a strong activity for reducing b-carotene bleaching

L41 :  significant DPPH free radical scavenging activity  (78.4%) at high concentration (1000 mg/ml)

L63 : antioxidative

L76 : (12.2%), camphene (10.4%) and borneol (3.1%)

L99 : effect on the normal cell line (HUVEC) …. lower than on the cancer cell lines (MCF-7, HepG2, LoVo)

L102 : oil on three cancer cell lines

L112 : apoptosis indicating

L113 : treatment with the essential oil causes induction of apoptosis.

L114 : late) increased proportionnally to the time of exposure

Figure 4 line 2 : cells, respectively.

L185 : were always about two-fold higher than that of the MIC’s.

L199 : dysplayed a great capacity to

L201 : significant radical scavenging activity (72 and

L209 : to be continued in order to find novel, more effective and affordable medications. With this aim,  searching for important and promising natural products from Saudi medicinal plants this investigation was undertaken.

L213 : Additionally we explored the anticancer, antimicrobial, antoxidant…as well as possible aoptotic induction  using

L217 : Apparenty, this is the first study

L222 : Our data showed that MBEO displays a high content of

L223 : (53.3%) is the predominant

L224 : This is to some extent in agreement

L228 : However, our results are not quite in line with a former report on the

L231 : Of course, the divergence in the

L232 : may be ascribed

L233 : time of harvest and extraction methods

L234 : plants, and thus influences greatly the chemical content of the oils

L236 : In the present study, we noticed

L240 : In agreement with our results

L241 :  ) on human colon tumor HT-29 cells .

L250 : Apoptotic cells display characteristic morphologies and partcular molecular features.

L252 : in anticancer chemotherapy.

L253 : The capacity to induce apoptosis is a solid marker for assessing potential agents fir the treatment of cancer (17,18). (delete : (Kerr et al., Taraphdar et al., 2001; ).

L259 : and can be visualized by the green fluorescence of the chromophor used (FITC/PI). (delete : (Alabsi et al., 2016; Zhang et al., 1997))

L260 : (19,20). DAPI dye was used for

L261 : (21). (delete : (Boutonnat et al., 1999)

L264 : These are indications of apoptosis (19, 22). (delete : (Alabsi et al., 2016; Jurisicova et al., 1996)).

L265 : suggest that MBEO can induce apoptsis in MMCF-7 cells.

L283 : February

L284 : , and a voucher sample

L290 : filtered

L299 : , respectively.

L303 : implemented

L327 : (HUVEC). Cels were grown in DMEM (without 2 mM 1-glutamine ?)

L328 : and incubated at

L341 : using a phase

L347 : ethanol

L352 : examined by EVOS

L356 : detection was performed according to

L359 : adherence

L361 : Then, the cells were incubated in … After incubation, 400

L362 : , and the samples were immediately

L374 : using the micro-well dilution assay

L376 : were prepared in suitable broth

L381, L394 : Then,

L412 : samples at 120 min, respectively.

L419 : To our knowledge, the present stud determined the

L423 : were predominant.

L423 : cearly showed that MBEO possesses cytotoxic,

L424 : sugges that MBEO can induce apoptosis (dependent on concentration ?)

L427 : antimicrobial effects.

Author Response

Reply to Reviewer 1:

The paper molecules-510419 « Analysis of chemical composition and assessment of cytotoxic, antimicrobial and antioxidant activities of the essential oil of Meriandra bergalensis growing in Saudi Arabia » by R.A. Monthana et al. provides new  interesting data on the components of the essential oil extracted from the aerial parts of this aromatic plant (M. bergalensis) of Saudi Arabia (MBEO), its antioxydant, antibacterial and antifungal and its cytotoxic activities on cancer cell lines such as MCF-7. Camphor (54.3%), 1,8-cineole (12,2%), camphene (10.4) and borneol (3.1%) are identified as the major compounds. The authors provide some evidence for the induction of antimicrobial effects an  cytotoxic effects on human cells in culture. Interestingly, the effects appear to be more pronounced on cancer than on normal cells. Moreover, MBEO is effective in β-carotene bleaching and DPPH free radical scavenging activity.

Thus, the paper is well presented and reflects genuine scientific work. However, unfortunately, in some parts it is lacking some precision and information (see General and specific comments). In its present form, it is not yet fully acceptable for publication. It should become fully acceptable when taking into account the reviewers’ comments.

Reply:  We thank the respected reviewer for this opinion and for all valuable comments and suggestions. We really  have benefited greatly from your feedback.

General comments

1.                  When talking about the cancer cell lines in chapter 2.2. it would give more impact to the results if the authors would shortly present them. (Human cancer cell lines MCF7 (breast cancer), LOVO (colon cancer) HepG2(liver cancer are compared to normal human umbilical vein endothelial (HUVEC) cells) ). It appears that with regard to cell viability changes, Hep62 cells are less sensitive to MBEO than the other two cancer cell lines. Is this statistically significative ? In other words, is the effect of MBEO specific for cancer cells and what cud be the reason for that?

Reply: We totally do agree with this comment. Actually we mentioned the types of the cancer cell lines in the section "Materials and methods" but because this section appears after the section "Results" was not first seen. It is now added to the section 2.2 in the revision.

From our previous investigations, we noticed that HepG2 cells are less sensitive to the most natural drugs investigated formerly including MBEO than the other two cancer cell lines in terms of IC50. The effect is not significantly different. The low sensitivity of the liver cells (HepG2) is maybe attributed to its tendency to develop multi-drug resistance.

2.        Figure 3 does not show necessarily apoptosis but mostly cell death.

Reply: It is true. There are two main types of cell death (apoptosis and necrosis) which occur independently, sequentially, as well as simultaneously and there is also an issue of distinguishing apoptosis from necrosis. Therefore, we carried out a further experiment using Annexin V-FITC/PI Apoptosis kit to confirm occurring of the apoptosis which can discriminate between early/late apoptosis vs necrosis.

3.     In Table 3, it looks as if  always the MBC or MBF values are values that are always about twice of those of the MIC values.  Is this just a question of the difference in methodology used ? The authors might want to comment on that.

Reply: Actually we don't have a precise reply why MBC or MFC values were about twice of those of the MIC values. These are the results what we obtained through our experiments. The method used was micro-broth dilution method (CLSI M07-A09), it is simple. Commonly the concentration which is obtained in MBC is higher than the concentration for MIC. To calculate the MBC, all of the clear wells (no turbidity) (Here we mean MIC and higher concentrations) have been cultivated on suitable microbial solid medium (agar plates further 24 h). Then, the lowest concentration of compound that inhibited bacterial growth was be considered as MBC.

4. The presentation is somewhat misleading when comparing mg/ml concentrations of MBEO with mg/ml concentrations for the standard drugs. The authors should make it very clear that MBEO (if the MBC and MIC values are statistically significant) may show an antibacterial actvity on Staphylococcus aureus that comes closest to that of the antibiotic gentamycin. What is so special with S. aureus ? Although somewhat reactive, all the other microbials appear to be very much more resistant to MBEO than to the standard test compounds. Does the growth medium used for MBEO treatment have a protective effect ?

Reply: We agree that the presentation was somewhat misleading, actually, the concentrations of standard antibiotics (µg/ml, it was not mg/ml) were less than all concentrations of the essential oil (mg/ml) for all tested microorganisms. So, the results showed that the biological activity of the essential oil as antimicrobial agent was less than that of standard antibiotics used in this work. The experiment was carried out without replicate, for this reason, the statistically significant differences have not been determined.

5.    L210 : It is not clear to the reader what the authors mean with « increasingly viable medications ».

Reply: We are totally in agreement with this comment. The sentence was rephrased in the revision.

6.   Lines 243-247 : The reasoning is not clear and not precise enough : Apparently, these sentences refer to  borneol, α-pinene and β-caryophyllene  and references 14-16 but not much to the present work. borneol, α-pinene and β-caryophyllene were found in small amounts (3.1, 2.5 and 0%, respectively). The types of essential oil and concentrations that affected the different cancer cell lines are not stated. Were these essential oil compounds predominantly present ? Do they allow the supposition that they may be mainly responsible for the observed cytoxic effects ? Or, is this meant to be a general statement that essential oils, somewhat similar to the camphor and 1,8-cineol types, show cytotoxic effects on cancer cells ? In any case, it is difficult to conclude because couriously enough, essential oil components when applied on their own have been shown in many instances to be less effective than the whole complex mixture of components that compose essential oils. Apparently, between individual essential oil components interesting interactions exist that underlie the overall biological activity of the essential oils. The authors may want to include a comment on this.

Reply:  This is a vital comment which we are totally in agreement with. It is true that that paragraph was a little confusing but we rephrased it in the revision to make our statement more obvious than before. Yes, as the respected reviewer said the references 14 to 16 are  referred to the compounds borneol, α-pinene and β-caryophyllene  which were previously investigated for their cytotoxicity and not to our predominant compounds in MBEO (Camphor and 1,8-Cineol). But as we mentioned before that it is only supposed that our major compounds in MBEO could be somehow contributed to the cytotoxic effect of MBEO along with other compounds in MBEO such as borneol, α-pinene which were previously found to be cytotoxic.

7.     Lines 248- 266 concerning apoptosis induction by MBEO : The authors should keep in mind that apoptosis induction is a complicated matter involving several different molecular pathways including mitochondrial functions. Also, the approach using cytofluorimetry is not an very easy matter (see Adan A, Alizada G, Kiraz Y, Baran Y, Nalbant A. Flow cytometry: basic principles and applications. Crit Rev Biotechnol. 2017 Mar;37(2):163-176. doi:10.3109/07388551.2015.1128876. Epub 2016 Jan 14. Review.). Very importanly, the measure of apoptosis provides indications of the underlying mechanisms. In thus regard, the observations reported by the authors are still preliminary but very promising. Thus, if possible, more research on this woud be worthwhile (see, for even more sophisticated approaches, Shaulov-Rotem Y, Merquiol E, Weiss-Sadan T, Moshel O, Salpeter S, Shabat D,Kaschani F, Kaiser M, Blum G. A novel quenched fluorescent activity-based probe reveals caspase-3 activity in the endoplasmic reticulum during apoptosis. Chem Sci. 2016 Feb 1;7(2):1322-1337. doi: 10.1039/c5sc03207e.) Monian P, Jiang X. Clearing the final hurdles to mitochondrial apoptosis:regulation post cytochrome C release. Exp Oncol. 2012 Oct;34(3):185-91.). By the way, also investigations on the concentration dependent induction of apoptosis (measuring different parameters of apoptosis such as morphological changes, DNA fragmentation (TUNEL), annexin, cytochrome C release, caspase 3 induction…) could constitute an interesting exension of the present work.

Reply: This is a very important comment which we are totally in agreement with. Thanks for the suggestions and the references. As the respected reviewer knows, flow cytometry is widely used for detection of apoptosis and considered as technology of choice in several studies and a majority of classical apoptotic characteristics can be quantitatively examined by flow cytometry according the following references:

1.      Vermes I, Haanen C, Reutelingsperger C. Flow cytometry of apoptotic cell death. J Immunol Methods. 2000 Sep 21;243(1-2):167-90. https://www.ncbi.nlm.nih.gov/pubmed/10986414 Cited by 804 - ‎Related articles

2.      Wlodkowic D, Skommer J, Darzynkiewicz Z. Flow cytometry-based apoptosis detection. Methods Mol Biol. 2009;559:19-32. https://www.ncbi.nlm.nih.gov/pubmed/19609746.

3.      Paul Allen and Derek Davies. Apoptosis Detection by Flow Cytometry. Flow Cytometry pp 147-163. https://link.springer.com/chapter/10.1007/978-1-59745-451-3_6.

As the respected reviewer said, apoptosis is a highly regulated process which involves different morphological and biochemical hallmarks and can be initiated through intrinsic (mitochondrial) or extrinsic pathways which induce cell death by activating caspases which can be detected by several methods as you explained. Unfortunately, all these methods requires a lot of time as well as purchase a lot of chemicals (primers, antibodies, kits) and it is difficult due so many issues in orders delay and insufficient fund. Definitely, this recommendation and suggestion will be taken in consideration in our coming experimentations and hope the coming manuscript will include more sophisticated experiments such western plot, DNA fragmentation, annexin, cytochrome C release, and caspase 3 induction.  

7.     L377 : 5%DMSO  may have on its own already a potent anti- oxidative (hydroxyl) radical effect. A control for DMSO alone is missing ? (the use of ethanol may give less biased results).

Reply: We also tested DMSO (even higher concentrations than 5%) on the microbial strains and no toxicity was observed on the microbial strains. Ethanol is toxic for the microbial strains therefore we did not use it in our experiments.

8.     L376 : dilution of MBEO in broth may reduce the actual activity of MBEO. Did the authors check for that ? The growth medium may reduce the possibility of MBEO to interact with bacterial or fungal cells. (This could explain the observed relative resistance  of most microbials to MBEO).

Reply: We mixed the oil (MBEO) with DMSO to overcome this problem and to get a good mixing of the oil with the broth media.  The test was performed according to the methods for standard dilution antimicrobial susceptibility tests. The growth medium was applied without oil as control and no inhibiting activity against tested microbes was observed. Our results also confirmed that the growth medium did not reduce the interaction possibility of the standard antibiotics with bacterial or fungal cells. Maybe further studies are needed to confirm the effect of growth medium on the activity of MBEO. This concern will be taken in consideration in the future in our coming experiments with natural drugs.

Specific comments

 L34 : , especially Reply: corrected accordingly.

L36 : quantitatively confirmed for MCF-7 cells. Reply: corrected accordingly.

L40 : a strong activity for reducing b-carotene bleaching Reply: corrected accordingly.

L41 :  significant DPPH free radical scavenging activity  (78.4%) at high concentration (1000 mg/ml) Reply: corrected accordingly.

L63 : antioxidative. Reply: corrected accordingly.

L76 : (12.2%), camphene (10.4%) and borneol (3.1%). Reply: corrected accordingly.

L99 : effect on the normal cell line (HUVEC) …. lower than on the cancer cell lines (MCF-7, HepG2, LoVo). Reply: corrected accordingly.

L102 : oil on three cancer cell lines. Reply: corrected accordingly.

L112 : apoptosis indicating. Reply: corrected accordingly.

L113 : treatment with the essential oil causes induction of apoptosis. Reply: corrected accordingly.

L114 : late) increased proportionnally to the time of exposure. Reply: corrected accordingly.

Figure 4 line 2 : cells, respectively. Reply: corrected accordingly.

L185 : were always about two-fold higher than that of the MIC’s. Reply: corrected accordingly.

L199 : dysplayed a great capacity to. Reply: corrected accordingly.

L201 : significant radical scavenging activity (72 and…. Reply: corrected accordingly.

L209 : to be continued in order to find novel, more effective and affordable medications. With this aim,  searching for important and promising natural products from Saudi medicinal plants this investigation was undertaken. Reply: The sentence was rephrased correctly.

L213 : Additionally we explored the anticancer, antimicrobial, antoxidant…as well as possible aoptotic induction  using. Reply: corrected accordingly.

L217 : Apparenty, this is the first study. Reply: corrected accordingly.

L222 : Our data showed that MBEO displays a high content of. Reply: corrected accordingly.

L223 : (53.3%) is the predominant. Reply: corrected accordingly.

L224 : This is to some extent in agreement. Reply: corrected accordingly.

L228 : However, our results are not quite in line with a former report on the. Reply: corrected accordingly.

L231 : Of course, the divergence in the…. Reply: corrected accordingly.

L232 : may be ascribed. Reply: corrected accordingly.

L233 : time of harvest and extraction methods. Reply: corrected accordingly.

L234 : plants, and thus influences greatly the chemical content of the oils. Reply: corrected accordingly.

L236 : In the present study, we noticed. Reply: corrected accordingly.

L240 : In agreement with our results. Reply: corrected accordingly.

L241 :  ) on human colon tumor HT-29 cells. Reply: corrected accordingly.

L250 : Apoptotic cells display characteristic morphologies and partcular molecular features. Reply: corrected accordingly.

L252 : in anticancer chemotherapy. Reply: corrected accordingly.

L253 : The capacity to induce apoptosis is a solid marker for assessing potential agents fir the treatment of cancer (17,18). (delete : (Kerr et al., Taraphdar et al., 2001; ). Reply: deleted accordingly.

L259 : and can be visualized by the green fluorescence of the chromophor used (FITC/PI). (delete : (Alabsi et al., 2016; Zhang et al., 1997). Reply: deleted accordingly.

L260 : (19,20). DAPI dye was used for. Reply: corrected accordingly.

L261 : (21). (delete : (Boutonnat et al., 1999). Reply: deleted accordingly.

L264 : These are indications of apoptosis (19, 22). (delete : (Alabsi et al., 2016; Jurisicova et al., 1996)). Reply: deleted accordingly.

L265 : suggest that MBEO can induce apoptsis in MMCF-7 cells. Reply: corrected accordingly.

L283 : February. Reply: corrected accordingly.

L284 : , and a voucher sample. Reply: corrected accordingly.

L290 : filtered. Reply: corrected accordingly.

L299 : , respectively. Reply: corrected accordingly.

L303 : implemented. Reply: corrected accordingly.

L327 : (HUVEC). Cels were grown in DMEM (without 2 mM 1-glutamine ?) Reply: corrected accordingly.

L328 : and incubated at. Reply: corrected accordingly.

L341 : using a phase. Reply: corrected accordingly.

L347 : ethanol Reply: corrected accordingly.

L352 : examined by EVOS. Reply: corrected accordingly.

L356 : detection was performed according to. Reply: corrected accordingly.

L359 : adherence. Reply: corrected accordingly.

L361 : Then, the cells were incubated in … After incubation, 400. Reply: corrected accordingly.

L362 : , and the samples were immediately. Reply: corrected accordingly.

L374 : using the micro-well dilution assay. Reply: corrected accordingly.

L376 : were prepared in suitable broth. Reply: corrected accordingly.

L381, L394 : Then, Reply: corrected accordingly.

L412 : samples at 120 min, respectively. Reply: corrected accordingly.

L419 : To our knowledge, the present stud determined the. Reply: corrected accordingly.

L423 : were predominant. Reply: corrected accordingly.

L423 : cearly showed that MBEO possesses cytotoxic, Reply: corrected accordingly.

L424 : sugges that MBEO can induce apoptosis (dependent on concentration ?) Reply: corrected accordingly.

L427 : antimicrobial effects. Reply: corrected accordingly.

Reviewer 2 Report

This is an interesting paper dealing chemical composition and assessment of cytotoxic, antimicrobial and antioxidant activities of the essential oil of Meriandra bengalensis growing in Saudi Arabia. However, it is necessary to correct:

1) Improve introduction: discussing about volatile chemical composition in the genus Meriandra, identification process and other informations;

2) Line 43: Choose keywords that are not in the article title;

3) Line 32. Insert point (.) after (10.4%);

4) Line 79 (Table 1): Insert RI (Retention Index) from the literature;

5) Correct the word 1,8-cineol for 1,8-cineole;

6) It’s not possible confirm that camphor and 1,8-cineole is responsible for the great cytotoxic activity of the MBEO. The authors should evaluate the activity of these compounds together or alone;

7) Authors need to review the text writing.

Author Response

Reply to Reviewer 2:

This is an interesting paper dealing chemical composition and assessment of cytotoxic, antimicrobial and antioxidant activities of the essential oil of Meriandra bengalensis growing in Saudi Arabia. However, it is necessary to correct:

Thanks a lot for the respected reviewer for the valuable comments and suggestions.

1) Improve introduction: discussing about volatile chemical composition in the genus Meriandra, identification process and other informations;

Reply: discussing about volatile chemical composition in the genus Meriandra is mentioned in details in our manuscript. Indeed we chose to discuss the chemical composition of the Meriandra plants in the section (Discussion/lines:222-242) instead of the introduction. What of information was provided in the section "introduction" was all what could be found about the Meriandra plants. As we mentioned in the Introduction the genus Meriandra (family: Lamiaceae) is represented by only four species. Thus, the existing knowledge was really limited.

2) Line 43: Choose keywords that are not in the article title

Reply: we changed and added key words which are not in the title

3) Line 32. Insert point (.) after (10.4%)

Reply: it is corrected accordingly

4) Line 79 (Table 1): Insert RI (Retention Index) from the literature

Reply: it is really difficult to take the RI values from literature since different columns in GC and GC/MS experiments are used in several study and consequently the values are not comparable. In fact, we believe that our  Library (Baser Library of volatile oil constituents) which was established through several years of work on volatile oils is a very good Library on the basis of their retention indices.

5) Correct the word 1,8-cineol for 1,8-cineole

Reply: it is corrected accordingly

It’s not possible confirm that camphor and 1,8-cineole is responsible for the great cytotoxic activity of the MBEO. The authors should evaluate the activity of these compounds together or alone.

Reply: This is a very important comment which we are totally in agreement with. It is true that that paragraph was a little confusing but we rephrased it in the revision to make our statement more obvious than before. As we mentioned before that it is only supposed that our major compounds in MBEO e.g. camphor and 1,8-cineole could be somehow contributed to the cytotoxic effect of MBEO along with other compounds in MBEO such as borneol, α-pinene which were previously found to be cytotoxic. It is true that it is better to test these compounds on the cancer cells however; we did not isolate them and do not have them available as reference compounds in our lab.

7) Authors need to review the text writing

Reply: Carefully the manuscript was read again and several corrections were made to put the manuscript in a better shape.

Reviewer 3 Report

The Authors present very preliminary data not sufficiently supported by experiments. Therefore a clear design of the experiments is suggested.

Author Response

Reply to Reviewer 3:

Comments and Suggestions for Authors

The Authors present very preliminary data not sufficiently supported by experiments. Therefore a clear design of the experiments is suggested.

Reply: We respect the comment of the third reviewer although we did not understand what is requested from us.

Actually there are a lot of research groups involved in searching of essential oils and their pharmacological activities. Moreover, several international journals with high impact factors are interested in publishing such works. Experiments on volatile oils are carried out with GC and GC/MS experiments and this is what was performed in our labs. In addition, we evaluated three different pharmacological activities using standard methods. So, our study was indeed well-designed. It is true that it was better if the apoptotic pathway was deeply investigated and clarified however; as we mentioned above that these methods requires a lot of time as well as purchase a lot of chemicals (primers, antibodies, kits) and it is difficult due so many issues in orders delay and insufficient fund. Definitely, this recommendation and suggestion will be taken in consideration in our coming experimentations and hope the coming manuscript will include more sophisticated experiments such western plot, DNA fragmentation, annexin, cytochrome C release, and caspase 3 induction.  

Round 2

Reviewer 3 Report

The MS, as it is, needs to be improved. The MS shows very preliminary data. The reviewer strongly suggests  to include signal transduction study, molecular apoptosis analysis, proliferation study by single point assay in IF. Reviewer's job is to understand  the general story supported by  data info whether they give enough novelty in order to be published. 

Author Response

Reply to Reviewer 3:

Comments and Suggestions for Authors

The MS, as it is, needs to be improved. The MS shows very preliminary data. The reviewer strongly suggests  to include signal transduction study, molecular apoptosis analysis, proliferation study by single point assay in IF. Reviewer's job is to understand  the general story supported by  data info whether they give enough novelty in order to be published.

Reply: We thank the respected reviewer for the comments. We would like to explain something regarding the manuscript. The plant Meriandra bengalensis contains mainly two types of constituents namely diterpenoids and volatile oils. Few years ago, we investigated the crude extract and isolated few diterpenoids (Tanshinons) which showed promising cytotoxic and antimicrobial activities. Recently we had the following question: could the volatile oil-part also be responsible for the previously found cytotoxic and antimicrobial activities. Therefore; the research design of this study was to carry out a phytochemical investigation using GC/FID and GC/MS of the pure volatile oil and to perform pharmacological investigation on cytotoxic and antimicrobial effects.  A lot of works on volatile oils which are published in several international journals, are like our manuscript (Phytochemical study and determination of one or two effects). Since one of the biochemists (Dr. Fahd Nasr) joined recently to our team, he encouraged us to go deeper with molecular apoptosis analysis. Thus, using labelling with annexin VFITC and propidium iodide (PI) dyes and flow cytometer analysis, the apoptosis induction was confirmed quantitatively against MCF-7 cells. These are the assays available in our lab at the this stage. However, we are intended to establish further experiments on proliferation, molecular apoptosis analysis and signal transduction study, e.g. western plot, DNA fragmentation, cytochrome C release, caspase 3 induction and etc…...  

Unfortunately, all these assays requires a lot of time as well as purchase a lot of materials (primers, antibodies, kits) and it is difficult due to many issues like orders delay and insufficient fund to perform them this time. Definitely, this recommendation and suggestion will be taken in consideration in our coming experimentations and hope the coming manuscript will include more such sophisticated experiments. Thus, we hope that the respected reviewer accept our apology for our inability to perform these tests and for the lack of facilities at this time.

We are totally in agreement with the respected reviewer that further experiments are needed to explain the full mechanism of apoptosis. However, if the respected reviewer allows us we think that the manuscript shows advanced and promising results.

We thank in advance the respected reviewer for the understanding the difficulties that researchers are facing.

 Kind regards